# Prevalence of Non-Toxigenic *Clostridioides difficile* in Diarrhoea Patients and Their Clinical Characteristics

**DOI:** 10.3390/antibiotics12091360

**Published:** 2023-08-24

**Authors:** Cheon-Hoo Jeon, Si-Ho Kim, Yu Mi Wi

**Affiliations:** Division of Infectious Diseases, Samsung Changwon Hospital, Sungkyunkwan University School of Medicine, Changwon 51353, Republic of Korea; cheonhoo144.jun@samsung.com (C.-H.J.); siho.kim@samsung.com (S.-H.K.)

**Keywords:** *Clostridioides difficile*, non-toxigenic *Clostridioides difficile*, toxigenic *Clostridioides difficile*, diabetes, proton pump inhibitors, glycopeptides

## Abstract

Non-toxigenic *Clostridioides difficile* (NTCD) has been shown to decrease the risk of recurrent *C. difficile* infection (CDI) in patients following metronidazole or vancomycin treatment for CDI. Limited data on the prevalence of NTCD strains in symptomatic patients and their clinical characteristics are available. We conducted this study to investigate the prevalence of NTCD in diarrhoea patients and their clinical characteristics. Between July 2017 and June 2018, unduplicated stool specimens were collected from patients with diarrhoea. The characteristics and episodes of *C. difficile* infection in patients with NTCD and toxigenic strains were compared. Among the 1182 stool specimens collected, 236 (18.5%) were identified as growing *C. difficile*, and 19.5% of the identified isolates were found to be NTCD. Multivariate analysis showed that community-onset diarrhoea (OR = 4.13, 95% CI 1.07–15.97; *p* = 0.040), underlying diabetes (OR = 3.64, 95% CI 1.46–9.25; *p* = 0.006), previous use of glycopeptides (OR = 4.75, 95% CI 1.37–16.42; *p* = 0.014), and the lack of use of proton pump inhibitors (PPIs) (OR = 3.57, 95% CI 1.39–9.09; *p* = 0.009) were independently associated with the NTCD group. Although there was no statistical significance, the number of CDI episodes occurring after 90 days tended to be lower in the NTCD group (2.2%) than in the toxigenic group (11.2%). A considerable portion of the *C. difficile* strains isolated from patients with diarrhoea showed NTCD. Further, more extensive studies are needed to clearly define the protective effects of NTCD strains in patients with diarrhoea.

## 1. Introduction

*Clostridioides difficile* is a Gram-positive, spore-forming anaerobic bacterium that accounts for 15–25% of all cases of antibiotic-associated diarrhoea [1]. *C. difficile* infection (CDI) remains an important cause of morbidity and mortality in healthcare-associated infections [2]. *C. difficile* produces toxins responsible for the disease, although not all strains produce toxins. Toxigenic *C. difficile* strains generally produce both toxins A (enterotoxin) and B (cytotoxin), and occasionally binary toxin (CDT); however, some are toxin A-negative due to mutations in the *tcdA* gene [3]. Ribotype (RT) 027 and other CDT-producing *C. difficile* strains have rarely been reported in Asian countries, whereas the most prevalent RTs are RT017, RT018, RT014, RT002, and RT001 [4,5]. No specific clinical features distinguish CDI from other causes of diarrhoea [6]. Therefore, rapid and accurate diagnosis of CDI is essential for the initiation of appropriate antibiotics and for controlling its spread [7]. The widely used assays are *C. difficile* toxin A and B enzyme immunoassays (toxin EIA), which detect free toxins in faeces; glutamate dehydrogenase (GDH) tests, which detect a common antigen produced by *C. difficile*; and nucleic acid amplification tests (NAATs), which detect toxin genes [6,8,9]. Toxin EIA correlates better with disease than GDH or NAAT, but this method has poor sensitivity, leading to missed cases [10,11]. In contrast, NAATs cannot differentiate between active infection and asymptomatic carriage because they detect toxin genes alone but not toxin production [11,12,13,14].

In contrast, non-toxigenic *C. difficile* (NTCD) strains cannot produce toxins and are usually not associated with symptomatic infections. NTCD strains do not contain the pathogenic island encoding toxins A and B (PaLoc genes) or exhibit dysregulation of the *tcdA* and *tcdB* genes, and they do not produce enough toxins to cause disease [15,16,17]. NTCD gastrointestinal colonisation has been shown to prevent CDI through exposure to a toxigenic strain [18,19]. NTCD-M3 has also been shown to be effective in the prevention of recurrent CDI in patients who have been treated with metronidazole or vancomycin for CDI [18]. Therefore, a promising preventive measure against CDI is the use of NTCD to colonise the destroyed gut after antibiotic treatment and prevent colonisation by toxigenic *C. difficile*.

Limited data on the prevalence of NTCD strains in symptomatic patients and their clinical characteristics are available. Therefore, we conducted a 1-year study of patients with diarrhoea to investigate the prevalence and of NTCD strains and patients’ characteristics.

## 2. Results

### 2.1. Characteristics of C. difficile Strains

Between July 2017 and June 2018, 1182 unduplicated specimens from 1858 stool specimens submitted for *C. difficile* toxin EIA testing were cultured. Among the 1182 specimens, 236 (18.5%) were positive for *C. difficile*. NTCD strains (n = 46) were observed in 19.5% of the *C. difficile* strains isolated from patients with diarrhoea. Among the 190 toxin gene polymerase chain reaction (PCR) + strains, 160 (84.2%) isolates were identified as A+B+CDT-, 22 (13.8%) as A-B+CDT-, and 8 (4.2%) as A+B+CDT+ strains. In addition, 83 isolates were identified as toxin EIA negative/toxin gene PCR positive and 107 isolates were identified as toxin EIA positive/toxin gene PCR positive (Figure 1 and Table 1).

### 2.2. Comparison of Clinical Features between Non-Toxigenic and Toxigenic C. difficile

The characteristics of patients with NTCD strains were compared with those with toxigenic *C. difficile* (Table 2). Community onset was more common (19.6% vs. 11.2%, *p* = 0.006), and underlying diabetes was more prevalent in patients with NTCD strains (34.8% vs. 17.8%, *p* = 0.022). Patients with NTCD strains were less likely to have received antimicrobial therapy in the preceding month (78.3% vs. 90.6%, *p* = 0.039) but were more likely to have received glycopeptide therapy (18.2% vs. 5.9%, *p* = 0.031). The use of proton pump inhibitor (PPI) therapy was significantly lower in patients with NTCD strains than in patients with toxigenic *C. difficile* (21.7% vs. 40.6%, *p* = 0.025).

### 2.3. Predictors of NTCD Strains in Patients with Diarrhoea

A multivariate analysis of the potential predictors associated with NTCD strains is shown in Table 3. Variables with a *p*-value < 0.05 in the univariate analysis were included in the subsequent multivariate analysis. A logistic regression model revealed that community onset [OR = 4.13, 95% CI 1.07–15.97; *p* = 0.040)], underlying diabetes (OR = 3.64, 95% CI 1.46–9.25; *p* = 0.006), vancomycin therapy in the preceding month (OR = 4.75, 95% CI 1.37–16.42; *p* = 0.014), and non-concurrent use of PPIs (OR = 3.57, 95% CI 1.39–9.09; *p* = 0.009) were independent predictors of NTCD strains in patients with diarrhoea.

### 2.4. Comparison of Clinical Signs and Subsequent CDI Episodes between Non-Toxigenic and Toxigenic C. difficile

Clinical signs were similar between non-toxigenic and toxigenic *C. difficile*. A body temperature > 38.0 °C and a white blood cell count of >15,000/µL were more common in those with toxigenic *C. difficile*. The number of subsequent CDI episodes following a period of 90 d tended to be lower in the NTCD group (2.2%) than in the toxigenic group (11.2%), although there was no statistical significance (Table 4).

## 3. Discussion

In this study, we investigated the prevalence and characteristics of NTCD in patients with diarrhoea. NTCD strains were observed in 19.5% of *C. difficile* strains isolated from patients with diarrhoea. The clinical features of patients with NTCD were significantly different from those with toxigenic *C. difficile*. NTCD was associated with community-onset diarrhoea, underlying diabetes, previous use of glycopeptides within 1 month, and non-concurrent use of PPIs. Although the difference was not statistically significant with this sample size, the amount of subsequent CDI episodes after 90 days tended to be lower in the NTCD group (2.2%) than in the toxigenic group (11.2%).

Little is known about the prevalence of non-toxigenic strains. A few studies on the epidemiology of NTCD strains among hospitalised patients with diarrhoea have been published in Asia [20,21,22]. A previous study in Malaysia showed that the prevalence of non-toxigenic strains was 12.4% (54/437) in inpatients aged 18–80 years who experienced diarrhoea [20], which was comparable to the reported prevalence in Indonesia (10.6%) [21] and Thailand (15.6%) [22]. While further extensive research is needed to determine the protective effects of NTCD strains in patients with diarrhoea, anecdotal reports and a recent multinational study indicate that in Southeast Asia, where the prevalence of NTCD is high, CDI generally presents as self-limiting diarrhoea, and recurrence is rare [23]. The prevalence of NTCD strains in our study (19.5%) was higher than that reported in previous studies [20,21,22], which could be due to differences related to the characteristics of patient populations, antimicrobial stewardship practices, or local animal or environmental reservoirs of *C. difficile*. Intriguingly, the amount of subsequent CDI episodes after 90 days tended to be lower in the NTCD group (2.2%) than in the toxigenic group (11.2%). This may be due to the protective role of non-toxigenic *C. difficile*. One study found that 88 (46%) of 192 asymptomatic patients with *C. difficile* colonisation carried NTCD, and the colonisation with *C. difficile* was associated with lower CDI rates than those who were not colonised [19]. In addition, NTCD-M3 has been demonstrated to be effective in the prevention of recurrent CDI in a Phase 2 clinical trial, where vancomycin and metronidazole were the treatment antibiotics [18]. Based on these data, NTCD-M3 is currently being developed as a novel live biotherapeutic to reduce recurrent CDI. A recent study also demonstrated the ability of NTCD-M3 to colonise hamsters after treatment with either fidaxomicin or vancomycin [24]. Therefore, a Phase 3 clinical trial is currently being developed to confirm this effect after treatment with current standard of care antibiotics, which will likely include fidaxomicin as well as vancomycin.

A total of 83 isolates (43.7%) among the 190 isolates with a positive toxin gene PCR test were shown to be toxin EIA negative. This finding is consistent with a previous study, which found that 55.3% (162 of 293) of patients with a positive *C. difficile* PCR test result lacked toxin using the clinical toxin immunoassay test [11]. Guerrero D.M. et al. also demonstrated that 43 (32%) patients who underwent EIA tested negative for toxin in a prospective study of 132 patients with a diagnosis of CDI using PCR [25]. Several studies provide evidence that *C. difficile* colonisation is frequently observed in hospitalised patients, and many cases of nosocomial diarrhoea are of non-infectious origin. In addition, it is possible that clinical toxin tests can miss the presence of toxin at low concentrations [26,27].

The genes *tcdA* and *tcdB* that encode toxins A and B, respectively, are located near other genes that control their expression (*tcdC* and *tcdR*) and the release of biologically active forms of the toxin (*tcdE*). This region of the *C. difficile* genome is referred to as the pathogenicity locus (PaLoc). NTCD isolates lack an intact PaLoc, which means they do not produce toxin A or B and are typically not associated with symptomatic infections [15,16,28]. In addition to these strains, some NTCD strains have altered regulation of the *tcdA* and *tcdB* genes, resulting in insufficient expression of the bioactive toxins that cause disease. These strains are considered ‘phenotypically’ non-toxigenic. For instance, the M90 strain carries a PaLoc but fails to produce detectable toxin levels, possibly due to poor gene transcription [29]. In our study, 83 isolates were toxin EIA negative/toxin gene PCR positive, which could be ‘phenotypically’ non-toxigenic or ‘genotypically’ toxigenic. Toxin EIA positivity best defines true cases of *C. difficile* infections. Therefore, we excluded 83 toxin EIA negative/toxin gene PCR positive strains from the study and compared the NTCD strain and toxin-producing toxigenic *C. difficile* groups to evaluate the characteristics of NTCD in patients with diarrhoea. NTCD was associated with community-onset diarrhoea, underlying diabetes, previous use of glycopeptides within 1 month, and the lack of concurrent use of PPIs in our study. Diabetes increases the risk of recurrent CDI; however, metformin treatment seems to have a protective effect against the development of CDI by altering the gut microbiota composition [30]. Our study also showed that the proportion receiving metformin was higher in the NTCD group than in the toxigenic group (37.0% vs. 12.1%, *p* = 0.001). On the other hand,, previous use of glycopeptides within 1 month and no concurrent use of PPIs were also more frequent in the NCTD group. Approximately 80–90% of each intravenous vancomycin dose is excreted in urine and has little effect on intestinal microbiota, and intravenous vancomycin is classified as a low-risk antibiotic for CDI. In addition, pooled analysis of 50 studies showed a significant association between PPI use and risk of developing CDI (OR = 1.26, 95% CI, 1.12–1.39) compared with non-users [31]. Although the mechanism underlying the association between the aforementioned factors and NTCD has not been investigated, these factors may not affect the gut microbiota, including NTCD, allowing NTCD to retain its protective role against CDI.

The present study had some limitations. First, all the strains included in this study were clinical isolates from patients with diarrhoea, but NTCD-positive patients may still be carriers, and their diarrhoea is probably due to an alternative aetiology. In addition, NTCD may be involved in mixed infections with toxigenic strains. Second, environmental contamination by *C. difficile* could be an important source of transmission [32]. Therefore, subsequent CDI development may be affected by differences in cleaning and disinfection practices. In a study of a large patient cohort using whole-genome sequencing, the researchers were able to determine an association with previous CDI cases in only 55% of newly developed cases [33]. Third, although the value of leukocytes in the faeces of patients with diarrhoea is considered significant, we did not measure leukocyte counts in the faeces. Finally, this study was conducted at a single centre and had a small sample size. Subsequent CDI episodes after 90 days exhibited differences between the NTCD group (2.2%) and the toxigenic group (11.2%); however, this difference did not reach statistical significance. It is important to acknowledge that a small sample size can often yield statistically insignificant results. This is due to the increased variability within the data associated with limited sample sizes, resulting in wider confidence intervals and larger *p*-values. Consequently, even if a true difference or effect exists within the population, a small sample size may not provide sufficient evidence to establish its statistical significance. In addition, while the odds ratios have statistically significant *p*-values (*p* < 0.05), the confidence intervals still encompass a broad range of values. The calculated odds ratios provide valuable insights, but the wide confidence intervals caution us to interpret the results cautiously and with awareness of the potential variability and uncertainty in the estimates. Therefore, the results need to be validated through a relatively larger study or a more refined study design, in which an appropriate sample size is calculated beforehand, particularly focusing on providing more detailed information on the characteristics of patients with diarrhoea.

## 4. Materials and Methods

Study Design and Population

This cohort study was conducted among patients hospitalised at Samsung Changwon Hospital between July 2017 and June 2018. Eligible specimens were prospectively identified by reviewing the stool specimens submitted for toxin EIA during the study period. Only unduplicated specimens from patients with at least three loose or watery stools within 24 h were included. Clinical data were retrospectively obtained from the medical records of each patient.

Laboratory Testing

All stool samples were subjected to a *C. difficile* toxin EIA (RIDASCREEN 43 Clostridium difficile toxin A/B, R-Biopharm AG, Darmstadt, Germany) and reported clinically. Formed stools were rejected. Stool specimens were cultured anaerobically on *C. difficile* selective media (chromID *C. difficile*, bioMérieux, Lyon, France) for 48 h at 37 °C as previously described [34]. Putative *C. difficile* colonies were confirmed via colony analysis, odour, and Gram staining. These colonies were subcultured on a universal anaerobic culture medium (Brucella agar plate). DNA was extracted from the colonies grown on Brucella agar plates. *C. difficile* isolates were identified using 16S rRNA sequencing of the extracted DNA. PCR for *tcdA, tcdB, cdtA*, and *cdtB* genes was performed using the previously described method [3,35]. Two primer sets were used to detect the toxin A gene; primers NK3 and NK2 were derived from the nonrepeating portion of the *C. difficile* toxin A gene, and primers NK11 and NK9 were derived from the repeating portion of the *C. difficile* toxin A gene. A segment of the toxin B gene was amplified by using primer NK104 and primer NK105, which were derived from the nonrepeating sequence of the *C. difficile* toxin B gene. Probe NK106 was used and was 3′ end labelled with digoxigenin with a digoxigenin labelling kit. The thermal profile for primer pairs NK3-NK2 and NK104-NK105 was 35 cycles of 95 °C for 20 s, 55 °C for 120 s and 74 °C for 5 min. PCR amplification with primer pair NK11-NK9 was performed for 35 cycles, consisting of 95 °C for 20 s, 62 °C for 120 s and 74 °C for 5 min. Primers designed to amplify regions of *cdtA* and *cdtB* were as follows: cdtApos 5′-TGAACCTGGAAAAGGTGATG-3′ (position, cdtA 507–526); cdtArev 5′-AGGATTATTTACTGGACCATTTG-3′ (position, cdtA 882–860); cdtBpos 5′-CTTAATGCAAGTAAATACTGAG-3′ (position, cdtB 368–389); and cdtBrev 5′-AACGGATCTCTTGCTTCAGTC-3′ (position, cdtB 878–858). Reactions were subjected to 30 cycles of 94 °C for 45 s, 52 °C for 1 min and 72 °C for 1 min 20 s. The positive controls were ATCC 43594, 43598, and 9689, representing the A+B+CDT-, A-B+CDT-, and A+B+CDT+ ribotypes, respectively. Laboratory parameters were obtained 2 days before or 1 day after the diagnosis of diarrhoea. Albumin and CRP were measured as part of the automated chemistry analysis using a Roche Modular D2400 system (Roche Diagnostics, Indianapolis, IN), and the reference ranges of our institution are 3.1–5.2 g/dL for albumin and 0–5.0 mmol/L for CRP. White blood cell counts were obtained using a Sysmex XN-10 hematology analyzer.

Clinical Data Collection

Diarrhoea was defined as the passage of at least three loose or watery stools within 24 h. The case definition of CDI included patients with documented diarrhoea and a positive *C. difficile* EIA toxin assay or a positive toxin gene PCR. Clinical and laboratory characteristics, including age, sex, ward of acquisition, underlying comorbidities, recent medical history within 30 days of diarrhoea, concurrent infection, and concomitant medication were obtained from the medical records of each patient. To determine the severity of the illness, the modified Charlson’s comorbidity index was used for all patients [36]. We traced the development of a CDI patient over a 90-day period by reviewing medical records. For patients who were discharged within this timeframe, we confirmed the development of CDI through telephone records. A toxigenic *C. difficile* strain was defined as a case with positive PCR results for *tcdA, tcdB, cdtA*, or *cdtB* genes of an anaerobically cultured colony of the *C. difficile* strain with toxin production. An NTCD strain was defined as a case with negative toxin gene PCR results for *tcdA, tcdB, cdtA*, or *cdtB* genes of an anaerobically cultured colony of the *C. difficile* strain.

Statistical Analysis

The characteristics of NTCD and toxigenic *C. difficile* strains with toxin production were compared. The discrete data are expressed as frequencies and percentages, while continuous variables are presented as either mean ± standard deviation or median and interquartile range based on their distribution, which was determined using the Shapiro–Wilk normality test. Characteristics were compared using appropriate statistical tests such as the χ^2^ test, Fisher’s exact test, the two-sample *t*-test, or the Mann–Whitney U-test. A multivariable logistic regression model was employed to identify predictors of NTCD. In cases where the continuous data exhibited a skewed distribution, log transformations were performed during univariable analyses. Variables with a *p*-value < 0.10 in the bivariate analysis were considered candidates for multivariate analysis. The Hosmer–Lemeshow statistic was used to evaluate the goodness of fit of the final model. The Statistical Package for the Social Sciences for Windows (version 18.0; SPSS Inc., Chicago, IL, USA) was used to perform all analyses.

## 5. Conclusions

A considerable portion of *C. difficile* strains isolated from patients with diarrhoea showed NTCD. Community onset, underlying diabetes, previous use of glycopeptides, and non-concurrent usage of PPI were associated with NTCD strains. Further, more extensive studies are needed to clearly define the protective effects of NTCD strains in patients with diarrhoea.

## Figures and Tables

**Figure 1 antibiotics-12-01360-f001:**
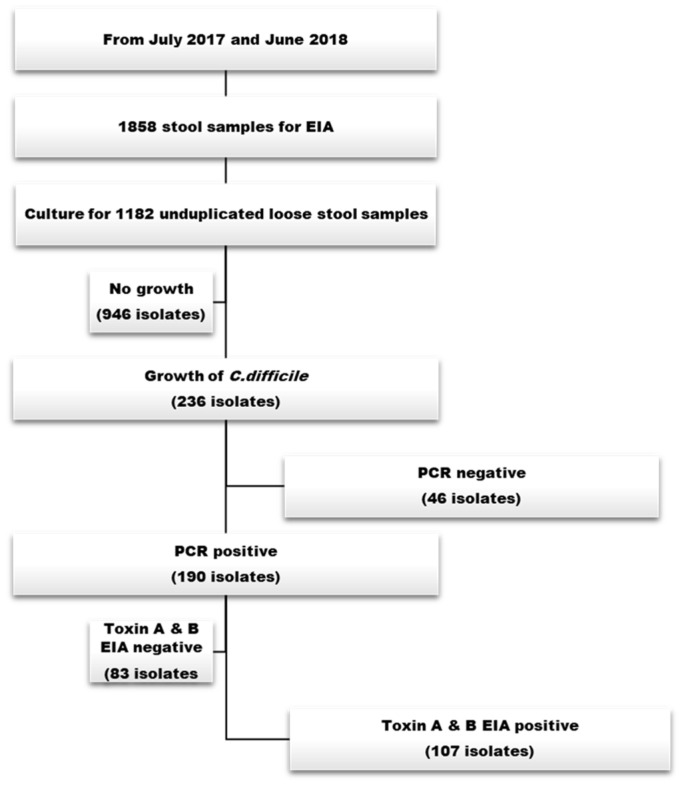
Overall study scheme.

**Table 1 antibiotics-12-01360-t001:** Distribution of toxin gene profiles in patients with diarrhoea.

Toxin Gene Profile (n = 236)	Toxin EIA Results	N (%)
A+B+CDT- strain	positive	92 (39.0)
	negative	68 (28.8)
A-B+CDT- strain	positive	11 (4.7)
	negative	11 (4.7)
A+B+CDT+ strain	positive	4 (1.7)
	negative	4 (1.7)
A-B-CDT- strain	negative	46 (19.5)

EIA—enzyme immunoassays.

**Table 2 antibiotics-12-01360-t002:** Comparison of characteristics between non-toxigenic and toxigenic *Clostridium difficile*.

	Non-Toxigenic (n = 46)	Toxigenic (n = 107)	*p*-Value
Age ≥ 65 y	23 (50.0)	68 (63.6)	0.117
Hospital stays, days, median (IQR)	7.5 (1.0–20.5)	7.5 (1.0–25.3)	0.562
ICU	9 (19.6)	12 (11.2)	0.169
Male sex	24 (52.2)	53 (49.5)	0.764
Category of infection			
Community onset	9 (19.6)	5 (4.7)	0.006
Community-onset healthcare facility associated	8 (17.4)	32 (29.9)	0.106
Hospital onset	29 (63.0)	70 (65.4)	0.778
Underlying disease			
Diabetes	16 (34.8)	19 (17.8)	0.022
Cerebrovascular disease	14 (30.4)	44 (41.1)	0.212
Cardiovascular disease	6 (13.0)	21 (19.6)	0.327
Chronic lung disease	5 (10.9)	9 (8.4)	0.760
Liver cirrhosis	2 (4.3)	3 (2.8)	0.637
Chronic renal disease without dialysis	8 (17.4)	12 (11.2)	0.299
Dialysis	4 (8.7)	7 (6.5)	0.735
Solid tumour	8 (17.4)	22 (20.6)	0.651
Solid organ transplantation	1 (2.2)	2 (1.9)	0.661
Charlson’s score, median (IQR)	2 (0–4)	2 (1–5)	0.175
Previous medical history within 1 month			
Operation	11 (23.9)	25 (23.4)	0.942
Immunosuppression	5 (10.9)	11 (10.3)	0.559
Antibiotic exposure	36 (78.3)	96 (90.6)	0.039
Extended spectrum cephalosporin	9 (20.5)	27 (26.5)	0.439
Quinolone	8 (18.2)	26 (25.5)	0.338
β-lactam/β-lactamases	8 (18.2)	20 (19.6)	0.841
Carbapenem	11 (25.0)	29 (28.4)	0.670
Glycopeptide	8 (18.2)	6 (5.9)	0.031
Gastrointestinal medication use at diagnosis			
PPI	10 (21.7)	43 (40.6)	0.025
H2 receptor antagonist	16 (34.8)	35 (33.0)	0.832
Probiotics	6 (13.0)	19 (17.9)	0.456
Concurrent systemic infection	27 (58.7)	51 (47.7)	0.211
Antibiotics use at the time of diagnosis	27 (58.7)	65 (60.7)	0.812

IQR—interquartile range; ICU—intensive care unit; PPI—proton pump inhibitor; Data are n (%) unless otherwise stated.

**Table 3 antibiotics-12-01360-t003:** Univariable and multivariable logistic regression analyses for independent predictors of non-toxigenic *C. difficile* among patients with diarrhoea.

	OR (95% CI)	*p*-Value	Adjusted OR (95% CI) ^a^	*p*-Value
Age ≥ 65 y	0.57 (0.29–1.15)	0.119		
* Hospital stays	1.02 (0.76–1.36)	0.903		
ICU	1.93 (0.75–4.95)	0.174		
Male sex	1.11 (0.56–2.22)	0.765		
Category of infection				
Community onset	4.96 (1.56–15.77)	0.007	4.13 (1.07–15.97)	0.040
Community-onset healthcare facility associated	0.49 (0.21–1.18)	0.110		
Hospital onset	0.90 (0.44–1.85)	0.778		
Underlying disease				
Diabetes	2.47 (1.13–5.41)	0.024	3.64 (1.46–9.25)	0.006
Cerebrovascular disease	0.63 (0.30–1.31)	0.213		
Cardiovascular disease	0.61 (0.23–1.64)	0.331		
Chronic lung disease	1.33 (0.42–4.20)	0.630		
Liver cirrhosis	1.58 (0.25–9.76)	0.625		
Chronic renal disease without dialysis	1.67 (0.63–4.40)	0.302		
Dialysis	1.36 (0.38–4.90)	0.637		
Solid tumour	0.81 (0.33–1.99)	0.651		
Solid organ transplantation	1.17 (0.10–13.20)	0.901		
Charlson’s score *	0.76 (0.42–1.35)	0.347		
Previous medical history within 1 month				
Operation	1.03 (0.46–2.32)	0.942		
Immunosuppression	1.06 (0.35–3.26)	0.913		
Antibiotic exposure	0.38 (0.14–0.98)	0.044		
Extended spectrum cephalosporin	0.71 (0.30–1.68)	0.440		
Quinolone	0.65 (0.27–1.58)	0.340		
β-lactam/β-lactamases	0.91 (0.37–2.26)	0.841		
Carbapenem	0.84 (0.38–1.88)	0.670		
Glycopeptide	3.56 (1.15–10.96)	0.027	4.75 (1.37–16.42)	0.014
Gastrointestinal medication use at diagnosis				
No PPI	0.41 (0.18–0.91)	0.028	0.28 (0.11–0.72)	0.009
H2 receptor antagonist	1.08 (0.52–2.24)	0.832		
Probiotics	0.69 (0.26–1.85)	0.458		
Concurrent systemic infection	1.56 (0.78–3.14)	0.212		
Antibiotics use at the time of diagnosis	0.92 (0.45–1.86)	0.812		

OR—odds ratio; CI—confidence interval; IQR—interquartile range; ICU—intensive care unit; PPI—proton pump inhibitor; * Log-transformation of the data is applied. ^a^ Variables with a *p*-value < 0.05 in the univariate analyses are included in the subsequent multivariate regression model. Hosmer–Lemeshow test, χ^2^ = 3.263, *p* = 0.917.

**Table 4 antibiotics-12-01360-t004:** Comparison of clinical signs and subsequent CDI episodes between non-toxigenic and toxigenic *C. difficile*.

	Non-Toxigenic (n = 46)	Toxigenic (n = 107)	*p*-Value
Signs at diagnosis			
Body temperature > 38.0 °C	14 (30.4)	51 (47.7)	0.048
Shock	1 (2.3)	9 (8.6)	0.282
Ileus	0 (0)	6 (5.8)	0.179
Laboratory finding			
White blood cell count > 15,000/µL	8 (18.6)	35 (35.4)	0.046
Acute kidney injury	2 (4.8)	6 (6.2)	0.545
Albumin level, g/dL, mean ± SD	3.5 ± 0.9	3.2 ± 0.7	0.073
CRP, mmol/L, mean ± SD	65.5 ± 78.7	71.7 ± 63.8	0.634
CDI development within 90 days	1 (2.2)	12 (11.2)	0.055

CRP—C-reactive protein; SD—standard deviation; CDI—*C. Difficile* infection; Data are n (%) unless otherwise stated.

## Data Availability

The data presented in this study are available on request from the corresponding author.

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
