# Peer review of "Prevalence of Non-Toxigenic Clostridioides difficile in Diarrhoea Patients and Their Clinical Characteristics"

_antibiotics, 2023, doi:10.3390/antibiotics12091360_

Round 1

Author Response

Dear Sir:

Editor

Antibiotics

August 19, 2023

I really appreciate you and reviewers for helpful suggestions and I feel that the quality of the manuscript has been significantly improved as a result. I provide point-by-point responses to the reviewers' comments. The text in bold signifies the comments made by a reviewer. The authors’ responses appear below each comment.

Modified portions were highlighted in yellow in the manuscript.

Sincerely,

___________________

Yu Mi Wi, M.D., PhD.

Division of Infectious Diseases, Samsung Changwon Hospital Sungkyunkwan University School of Medicine, 50 Hapseong-dong, Masanhoewon-gu, Changwon-si, Gyeongsangnam-do, Republic of Korea, 51353; Tel. number:82-55-290-6838;

Fax number:82-55-290-0041; E-mail address: yumi.wi@samsung.com

Reviewer 2 Report

The manuscript is very interesting and well presented.

I would have added the group of 83 EIA negative samples to the comparison with both EIA-positive group and the NTCD. As mentioned in lines 46-49, some NTCD may produce very low levels of toxins and the sensitivity of EIA is low, thus this 83 isolates may be also NTCD.

Author Response

I really appreciate you for your helpful suggestion. I agree with your opinion that some NTCD may produce very low levels of toxins and the sensitivity of EIA is low, thus this 83 isolates may be also NTCD. However, toxins produced by toxigenic strains may not be detected due to low sensitivity of toxin EIA test. I described the limitation of toxin EIA test as follows.

Line 42. Toxin EIA correlates better with disease than GDH or NAAT, but has poor sensitivity, leading to missed cases [10,11]. 

Line 158. In our study, 83 isolates were toxin EIA negative/toxin gene PCR positive, which could be ‘phenotypically’ non-toxigenic or ‘genotypically’ toxigenic. Toxin EIA positivity best defines true cases of C. difficile infections. Therefore, we excluded 83 toxin EIA negative/toxin gene PCR positive strains from the study and compared the NTCD strain and toxin-producing toxigenic C. difficile groups to evaluate the characteristics of NTCD in patients with diarrhoea.